# Association between Serum Lipids and Asthma in Adults—A Systematic Review

**DOI:** 10.3390/nu16132070

**Published:** 2024-06-28

**Authors:** Alexandra Maștaleru, Gabriela Popescu, Irina Mihaela Abdulan, Carmen Marinela Cumpăt, Alexandru Dan Costache, Cristina Grosu, Maria Magdalena Leon

**Affiliations:** 1Department of Medical Specialties I, “Grigore T. Popa” University of Medicine and Pharmacy, 700115 Iasi, Romania; alexandra.mastaleru@gmail.com (A.M.); irina.abdulan@yahoo.com (I.M.A.); adcostache@yahoo.com (A.D.C.); leon_mariamagdalena@yahoo.com (M.M.L.); 2Clinical Rehabilitation Hospital, 700661 Iasi, Romania; marinela.cumpat@umfiasi.ro (C.M.C.); fcristina_ro@yahoo.com (C.G.); 3Department of Medical Specialties III, “Grigore T. Popa” University of Medicine and Pharmacy, 700115 Iasi, Romania; 4Department of Neurology, “Grigore T. Popa” University of Medicine and Pharmacy, 700115 Iasi, Romania

**Keywords:** asthma, serum lipids, nutrition, HDL-cholesterol, LDL-cholesterol, triglycerides

## Abstract

(1) Background: Asthma is a syndrome found in both adults and children, characterized by airflow obstruction caused by the inflammation of the airways. In recent years, an increasing number of studies have found that lipid metabolism influences both the development and symptomatology of asthma. Lipid metabolism plays an important role both in the occurrence of exacerbations and in the reduction of lung inflammation. Our study aimed to identify any type of association between patients diagnosed with asthma and their serum lipids, including HDL-cholesterol, LDL-cholesterol, total cholesterol, and triglycerides in adults. (2) Methods: To find articles for our review, we searched two platforms: PubMed and Google Scholar. A total of 309 articles from two platforms were analyzed. Finally, 12 papers were selected from the initial pool of identified articles. (3) Results: The positive correlation between triglycerides, total cholesterol, low-density lipoprotein-cholesterol (LDL-cholesterol), and asthma has been demonstrated in several studies. Moreover, it appears that there is an association between biomarkers of type 2 inflammation and HDL and serum triglycerides in people with atopic status. Regarding the nutrition of asthmatic patients, the greatest impact on the development of the disease seems to be the consumption of fruit and vegetables. Several studies show that a predominantly vegan diet is associated with better control of the disease and a decrease in the number of pro-inflammatory cytokines. (4) Conclusions: Studies show a positive correlation between total cholesterol, triglyceride, and LDL-cholesterol levels and asthma and a negative correlation between HDL-cholesterol and asthma. Increased cholesterol values would lead to the stimulation of pro-inflammatory processes and the secretion of cytokines involved in these processes. The most successful diets for asthma patients seem to be those in which the consumption of fruit, vegetables, and high-fiber foods is increased because all of these food groups are rich in vitamins, antioxidants, and minerals.

## 1. Introduction

### Background/Rationale

Asthma (from the Greek ἅσθμα, ásthma, “choking”) is a common chronic inflammatory condition of the airways characterized by fluctuating and recurrent symptoms, reversible airflow obstruction, and bronchospasm. It is one of the most common chronic diseases worldwide and currently affects around 300 million people, with an additional 100 million people expected to be affected by 2025 [1,2]. In developing countries, the prevalence of asthma is increasing and is thought to be associated with urbanization. It seems that not so much genetic predisposition as environmental and lifestyle factors may be influencing the increased prevalence [2]. Asthma is now recognized as a heterogeneous disease, with an interaction between genetic and environmental factors [3,4].

Over the last decade and a half, there has been increasing discussion about the impact of diet on asthma prevalence. Studies show that people with low incomes are more prone to developing asthma, one of the important issues being that cheap foods are higher in unhealthy fats [5]. This dietary habit leads, over time, to dyslipidemia and thus to the development of a pro-inflammatory status [2].

In recent years, an increasing number of studies have found that lipid metabolism influences both the development and symptomatology of asthma. Obesity influences the onset of asthma exacerbations not only through anatomical changes due to the excess adipose tissue but also through chronic inflammation and abnormalities in metabolism [6,7].

Cholesterol is one of the main components in lung surfactant, the main source being plasma lipoproteins. HDL-cholesterol is also involved in the surfactant synthesis process, which additionally has anti-inflammatory and antioxidant roles. On the one hand, endothelial nitric oxide production is stimulated by HDL-cholesterol, and on the other hand, the production of pro-inflammatory cytokines is inhibited [8]. These associations can be observed in Figure 1.

Fatty acids regulate the phenotype and activate macrophages, thus increasing the level of inflammatory cytokines that play an important role in the pathogenesis of asthma. Moreover, fatty acids and cholesterol metabolism play a key role in the differentiation and activation of T cells. In addition, lipid molecules play a regulatory role in cellular processes affecting the evolution of allergic asthma, such as Th2 cell differentiation, the migration of eosinophils to the lungs, and IgE production. Lipids, especially extracellular lipids, regulate phagocytosis, secretion, and the development of alveolar macrophages. In obese patients with a high-fat diet, the number of lung macrophages significantly increases. Therefore, lipid metabolism plays an important role both in the occurrence of exacerbations and in the reduction of lung inflammation [7].

Our study aimed to identify any type of association between patients diagnosed with asthma and their serum lipids, including HDL-cholesterol, LDL-cholesterol, total cholesterol, and triglycerides in adults. Another aspect of interest was the study of factors that influence both asthma and serum lipids: eosinophil count, sleep quality, obesity, and metabolic syndrome.

## 2. Materials and Methods

### 2.1. Electronic Search Strategy

To find articles for our review, we searched two platforms: PubMed and Google Scholar. PubMed search results can be found in the table below. For Google Scholar, we used the same keywords as in PubMed but filtered the search by publication date, resulting in 44 articles. The results of the PubMed Electronic Search Strategy with PubMed Advanced Search Builder can be seen in Table 1.

In order to assess the quality of the studies included in our systematic review, we have performed a quality check using the risk of bias assessment tool. The results can be observed in Figure 2 and Figure 3 [9].

We have analyzed five domains in evaluating the risk of bias: bias arising from the randomization process, bias due to deviations from the intended intervention, bias due to missing outcome data, bias in measuring the outcome, and bias in selecting the reported result. None of the articles included in our systematic review had a high risk of bias, while some concerns were identified in 25% for the overall risk of bias.

### 2.2. Study Selection

A detailed step-by-step analysis of the articles was carried out by three independent reviewers (A.M., G.P., and I.M.A.) in duplicate. The main filters used in the search process were English language, publication period 2017–2024, and human participants over 18 years old. The search terms used were the association of the words asthma and total cholesterol, high-density lipoprotein, low-density lipoprotein, triglycerides, and serum lipids. Title assessment was the first step in the analysis process. Articles whose titles contained at least one of the search terms were included. Next, the type of paper was analyzed and posters, conference abstracts, presentations, book chapters, and animal studies were excluded. The three reviewers carefully read the abstracts of each article that met the above-mentioned criteria and selected for detailed analysis studies in which the included participants were diagnosed with asthma, and their lung function and serum lipids were assessed. Finally, a detailed analysis of the whole text was performed, and studies that did not show a correlation of any kind between serum lipids values and asthma were excluded. During the data abstraction phase, independent reviewers resolved any disagreements through discussion. If the three reviewers could not reach a consensus, a fourth reviewer (M.M.L.) was responsible for making the final decision. There were no specific limitations or requirements regarding the design of the studies included in the analysis, further details on study selection can be found in Figure 4.

## 3. Results

As keywords, we used the association between asthma and cholesterol, high-density lipoprotein, low-density lipoprotein, serum lipids, and triglycerides to identify articles on both platforms. Studies that enrolled human patients aged over 18 years, published between 2017 and 2024, were selected for analysis.

A total of 309 articles from two platforms were analyzed. The reviewers started by reading the abstracts to exclude articles that did not meet the eligibility criteria. Afterward, they analyzed the full text of the remaining articles. The reviewers removed any duplicates and ineligible articles. Finally, they selected 12 articles from the initial pool of identified articles.

Six of the selected studies focused more on the correlation between all components of the serum lipids and asthma. Following detailed analysis, we concluded that six showed a statistically significant result for the correlation between HDL-C, LDL-C, and asthma. The studies were carried out in several countries. Four of them took place in the USA, six of them in Asia (South Korea, India, Taiwan, Iraq, and China), and two in Europe (Italy, England, Scotland, and Wales). A total of 532,804 participants aged over 18 years were evaluated in total, five of the studies had participants with an average age over 40 years. In terms of the gender of the participants included, it appears that a higher percentage of women were enrolled. Patients enrolled in the studies analyzed had or were diagnosed with asthma during the evaluation. The correlation between serum lipids components and asthma severity and response to treatment was analyzed. A positive correlation was observed between TG, TC, LDL-C, and asthma. In addition, dyslipidemia has been associated with severe forms of the disease. Moreover, it was concluded that HDL-C would have a positive impact on the disease due to its antioxidant and anti-inflammatory effects.

A summarization of the main aspects of the studies included in the systematic review can be seen in Table 2.

Of the 12 articles included in the systematic review, only three of them discuss more extensively the impact of statins and corticosteroids on the lipid profile of asthmatic patients. However, after sensitivity analysis and variable correction, no statistically significant correlations were observed between serum lipids and patient medication. In three of the articles, it is mentioned that one of the exclusion criteria used was treatment with hypolipidemic drugs in patients. Discussions on the influence on serum lipids on both asthma treatment and associated pathologies are not found in the articles included in the study. Even though it is well known that beta blockers and ASA are not recommended in patients with asthma, according to specialty guidelines, none of the articles included in our systematic review evaluated this aspect.

## 4. Discussion

Cholesterol is responsible for many biochemical processes running in the human body. Once normal values are altered, mechanisms that lead to inflammation, and thus to the development of inflammation-based pathologies, including asthma, can be influenced. Over time, an association between cholesterol values and the risk of developing or worsening bronchial asthma has been observed. The positive correlation between triglycerides (TG), total cholesterol (TC), low-density lipoprotein-cholesterol (LDL-cholesterol), and asthma has also been demonstrated in a study of 500,000 people in England, Scotland, and Wales, which took place between 2006 and 2010. Of the total number of participants, 67,896 were diagnosed with asthma and 3071 were diagnosed with both asthma and atopic dermatitis. In addition to this negative association, a negative correlation was determined between high-density lipoprotein-cholesterol (HDL-cholesterol) levels and asthma risk [10].

In another study examining the relationship between serum lipids and asthma from a genetic point of view, it was shown that an increased level of total cholesterol and LDL-cholesterol leads to an increased risk of developing asthma. In addition, patients with genetically low HDL are in the same risk category as those with hypercholesterolemia [11].

Lei Liu and collaborators have highlighted through an observational cohort study, including 477 patients over the age of 18 diagnosed with asthma, that dyslipidemia appears to be associated with a severe form of asthma that is difficult to control therapeutically. Moreover, they observed more episodes of exacerbation with severe and moderate severity in patients with dyslipidemia than in those with normal serum lipids. Patients enrolled in the study were divided into two groups. One group consisted of 259 people with dyslipidemia, and one group of 218 people with normal serum lipids values. They were followed up for one year and evaluated by questionnaires, functional lung tests, and blood tests. The most common criterion for the diagnosis of dyslipidemia was abnormal total cholesterol values, found in 158 of the 259 patients. Following pulmonary function evaluation during the study, a higher number of exacerbation episodes was observed in patients with dyslipidemia than in those with normal serum lipids (12.1% vs. 6.6%). Further analysis showed that patients with elevated total cholesterol values were older, had a lower forced expiratory volume for 1 s and forced vital capacity ratio, and had a higher number of hospitalizations.

Regarding the asthmatic phenotypes, Liu et al. described the following associations: the non-allergic phenotype is associated with elevated total cholesterol values, the obesity-associated asthma type is correlated with high values of TG, and severe forms are associated with high values of LDL-C. Importantly, patients with dyslipidemia had lower IgE values and lower atopy rates than those with normal serum lipid values [12].

In a study published in 2024, Pratibha Singh and collaborators also attempted to assess the relationship between asthma and serum lipids in 107 people aged 18 years diagnosed with asthma. Patients included in the prospective observational cohort study were recruited over a period from 1 February 2021 to 31 January 2022. Lung function was assessed by spirometry, before and after bronchodilator administration, biological samples were taken, and patients were clinically examined after a complete history. Parameters used for serum lipids assessment were TC, LDL, HDL, TG, TC/HDL, TG/HDL, LDL/HDL, very low-density lipoproteins (VLDL), and non-HDL. After clinical and paraclinical evaluation, the patients were classified into two categories: patients with uncontrolled asthma (38 persons) and patients with controlled asthma (69 persons). After examining pulmonary function and serum lipids in each category of patients, it was concluded that the mean LDL value and TC/HDL ratio in those with uncontrolled asthma were significantly higher than in those with controlled asthma. Therefore, it is concluded that the values of these parameters may be correlated with asthma severity [13].

The authors of a 2017 systematic review focusing on the association between serum lipids and asthma prevalence explain that after a detailed analysis of the results obtained, it was observed that LDL-cholesterol and total cholesterol values were significantly higher among asthmatics compared to people included in the control group. Moreover, each component of the serum lipids may influence the occurrence of a particular form of asthma. As mentioned in many studies, HDL-cholesterol has a beneficial effect on asthma patients, decreasing allergen sensitivity and bronchial hyperresponsiveness, contrary to the effect of triglycerides, which have the opposite effect. However, HDL-cholesterol and triglyceride values could not be associated with asthma [22].

Dr. Vaibhav Sharawat et al. found no association between serum lipids and asthma severity. They recruited a total of 77 people over the age of 18 diagnosed with asthma and analyzed VLDL, total cholesterol, HDL-cholesterol, LDL-cholesterol, and triglyceride values. For more precise results, patients were grouped into severity groups: mild, moderate, and severe forms. The highest triglyceride and total cholesterol values were found in patients with a severe form of the disease. Even if no association was observed between the two factors studied, a correlation between asthma severity and triglyceride levels was observed in patients with normal body mass index [14].

### 4.1. LDL Cholesterol

Inflammation is involved in the pathogenesis of asthma, and LDL-cholesterol seems to play a role in this process. This particle transports cholesterol to peripheral tissues, which is important because an excess of this molecule can alter the function of the surfactant [23]. In addition, cholesterol plays an important role in type II cells, which has led many researchers to search for a link between statin treatment and the favorable outcome of asthma patients.

Low-density lipoproteins can be divided into three major classes: large-sized LDL, medium and small-dense LDL, or specific subclasses numbered one to seven. LDL1-2 are large particles with a low pro-inflammatory role, and LDL3-7 are small dense particles with an essential role in inflammation. Assuming that particles participating in the generation of atherosclerosis may also contribute to the triggering and maintenance of inflammatory processes in the airways, 70 asthma patients treated with inhaled corticosteroids were enrolled in an observational study. Ten healthy individuals were recruited as controls. Patients were classified according to the severity of the disease, according to GINA. Apparently, in those diagnosed with asthma, LDL-1 was lower than in healthy people, LDL-2 showed similar values, and LDL-3 was higher in asthmatics. As for the other subclasses, no significant data were obtained. Although in other studies a positive correlation was observed between LDL-1 and forced expiratory volume 1 (FEV1%) and a negative correlation between FEV1% and LDL-3, the statistical analysis concluded that there was no association between asthma severity and LDL-cholesterol subclasses [15].

However, it seems that in many related studies, it has been shown that among asthmatic patients, LDL-cholesterol had a much higher value than in healthy patients. The main protein of LDL-C is ApoB, playing an important role in lipid transport. In many studies, Apo A-I, the major structural protein of HDL-cholesterol, is positively correlated with asthma outcomes. In contrast, triglycerides, LDL-cholesterol, and ApoB are associated with unfavorable asthma outcomes.

One of the main factors in the pathophysiology of asthma is chronic airway inflammation. The inflammatory response is mediated by T helper 2 cells, which stimulate cytokine secretion, resulting in exaggerated airway inflammation. A panel of biomarkers identified in sputum, exhaled air, or blood samples is used to predict the type of clinical response to the inhaled treatment of asthma patients. Included in this panel is fractional exhaled nitric oxide (FeNO), determined by a simple and non-invasive test [16,24].

In the Department of Allergy and Pulmonology, Seoul St. Mary’s Hospital in the Catholic University of Korea, 167 asthma patients were enrolled to participate in a 9-month study. Blood samples were taken for serum lipids determination, including ApoA and ApoB values. In addition, pulmonary functional parameters and smoking status were assessed. The patients were divided into two groups according to the diagnostic method used: the group of patients diagnosed by pulmonary functional tests and symptoms, 93 participants, and the group of patients diagnosed by the specialist using symptomatology and clinical signs as criteria, 74 participants. Processing the accumulated data, the correlation between serum lipids and pulmonary functional parameters such as post-bronchodilator forced vital capacity (post-BD FVC), forced expiratory volume for 1 s (FEV1), FEV1/FVC ratio, and FeNO, was analyzed. Total cholesterol, LDL-cholesterol, and ApoB values showed a significant correlation with FeNO in patients diagnosed with asthma by functional tests, without a correlation in those diagnosed by the specialist. Even after adjusting for variables that might have influenced FeNO, a positive correlation between ApoB and FeNO was maintained. In addition, it appeared that patients with an increased ApoB level had lower post-BD FEV1 and post-BD FEV1/FVC values than those with normal levels. Cholesterol stimulates eosinophilic inflammation, leading to a predisposition to atopy, and the link between ApoB and FeNO found in this analysis may suggest an association between dyslipidemia and eosinophilic airway inflammation. Moreover, the correlation between ApoB and FeNO values brings to the fore the idea of using apolipoprotein B as a new marker in bronchial asthma [16].

### 4.2. HDL Cholesterol

HDL cholesterol is a group of lipoproteins that plays an important role in cholesterol metabolism. It also has anti-inflammatory, antithrombotic, and antioxidant properties that help maintain normal lung function.

The main structural protein of HDL-cholesterol, apolipoprotein A1 is associated with a favorable development of asthma. However, it appears that HDL-cholesterol may also have a pro-inflammatory role. When it binds to an acute-phase protein present even in asthma patients, called serum amyloid A (SAA), it can be converted into a dysfunctional particle with a pro-inflammatory role. The problem arises in asthma patients with high levels of SAA because it binds to HDL and stimulates inflammation [17].

From a large study database from 1999 to 2016, which included patients with and without a diagnosis of asthma, 146 patients with asthma and 154 people without a diagnosis of asthma were extracted. The aim was to quantify the level of SAA in each group. It was found that SAA levels were significantly higher in asthmatic patients than in healthy patients, with SAA values correlating with the severity of the condition, with higher levels most often found in patients with a severe form of the disease. Moreover, it appears that the endogenous form of HDL-cholesterol in the serum of asthmatics with high SAA levels showed a close association with this type of acute-phase protein having a high proinflammatory role on monocytes and neutrophils. Therefore, even though HDL-cholesterol is known to have an anti-inflammatory effect, improving lung function, in some asthmatic patients, it may be associated with SAA and transformed into a pro-inflammatory particle stimulating monocytes to secrete cytokines such as IL-6, TNF-α, and IL-1b [17].

### 4.3. Sleep Quality

Cardiovascular risk factors also seem to play an important role in sleep quality, with one of these factors being dyslipidemia. Several studies discuss the link between serum lipids and asthma, but a clear link between asthma and cholesterol categories has not yet been established.

A total of 1013 participants were included in a study conducted from 2004 to 2006. Blood tests were taken to determine their serum lipids and then to determine if there was a link between blood cholesterol levels, asthma, and sleep quality. Patients were divided into two groups, one consisting of 127 people with asthma and one consisting of 886 people without asthma [18]. The link between dyslipidemia and sleep quality in patients with obstructive sleep apnea has been discussed over time. It appears that hypoxia in this syndrome leads to altered lipid metabolism, thus patients have increased cholesterol levels [25].

In the study of 1013 patients, it was found that an increased HDL-cholesterol value is associated with better sleep quality in patients with asthma. One possible explanation is the anti-inflammatory and antioxidant effect of HDL-C, an important aspect in the pathophysiology of asthma. On the other hand, no association was found between LDL-cholesterol, total cholesterol, triglycerides, and sleep quality in sick patients. No link was observed between the value of any cholesterol category and sleep in healthy patients [18].

### 4.4. Blood Eosinophil Counts

Asthma can be classified into two types: eosinophilic and non-eosinophilic. In eosinophilic asthma, eosinophilic cells are the main participants in the inflammatory process, and their value correlates with the severity of the disease [19,26]. The National Health and Nutrition Examination Survey wanted to investigate whether there is a link between serum lipids and blood eosinophils in patients diagnosed with asthma. The study included 2544 patients over the age of 18 who had biological samples taken to determine their serum lipids and blood eosinophil count. Three univariable and multivariable weighted regression models were used to determine whether there was a significant correlation between the two factors in asthmatic patients. It was concluded that only HDL-cholesterol and not LDL-cholesterol, total cholesterol, and triglycerides correlated with the number of eosinophils in the blood. Several variables were adjusted for each model. For model I, no variables were adjusted; for model II, adjustments were made for age, sex, and race; for model III, adjustments were extended to sociodemographic aspects, comorbidities, medication, and diet. Eosinophil counts decreased by 49.07 U/L (model I), 45.68 U/L (model II), and 45.68 U/L (model III), respectively, for each additional unit of HDL-cholesterol (mmol/L). Therefore, an inverse and independent association between the two factors was observed [19].

The severity of asthma may be associated not only with the number of eosinophils but also with the serum periostin level. Increased periostin levels are found in asthmatic patients with type 2 inflammatory status. In a study involving 165 patients with atopic asthma and 163 patients without a diagnosis of asthma, an association between HDL-cholesterol levels and eosinophil counts was observed. Patients without a diagnosis of asthma were divided into two groups, one consisting of 79 people with non-asthmatic atopic status and one consisting of 84 people with non-atopic and non-asthmatic status. Biological samples were taken to determine the eosinophil count and serum lipids. Even though people with asthma had a higher total cholesterol and BMI than the other participants, after adjusting the quantified parameters, it was found that the number of eosinophils had a negative correlation with the amount of HDL-cholesterol and a positive correlation with the amount of triglycerides. Apparently, in the category of non-asthmatic patients, the number of eosinophils was also correlated with ApoA-I, not only with HDL-cholesterol. No association was identified between eosinophils and total cholesterol, LDL-cholesterol, apo B, and apo E in any category of patients. Further analysis using NMR spectroscopy showed this correlation not only with total HDL but also with some large HDL particles and periostin. To exclude possible effects of age, sex, race, BMI, and CRP from the results, a multivariable regression was performed using linear regression analysis, but the same results were obtained in the end. Therefore, it appears that there is an association between biomarkers of type 2 inflammation and HDL and serum triglycerides in people with atopic status. Some studies suggest that HDL-cholesterol-associated proteins rather than cholesterol particles may be involved in this process [20].

### 4.5. Obesity and Metabolic Syndrome

Obesity and overweight are risk factors for asthmatic patients and are a predisposing factor. Obesity leads to changes in serum lipids, increasing insulin resistance and promoting systemic inflammation [27,28]. In patients with excessive body weight, lung function and volume are also reduced due to the pressure exerted by excess intra- and extra-thoracic and abdominal fat deposits [29]. All of this creates a disastrous terrain for the asthmatic patient, leading to severe respiratory symptoms and a decline in lung function [30]. Moreover, dyslipidemia is additionally a risk factor for the development of cardiovascular disease, leading to a poor prognosis among asthmatics.

To observe the link between overweight and asthmatic patients, we analyzed a case–control study that ran from 2010 to 2012 and included 348 people with a mean age of 34.34 ± 11.58 years. This study included 190 people with asthma, 48 healthy people to form the control group, and 110 patients diagnosed with allergic rhinitis, whom we excluded from our analysis. Asthma patients were divided into three groups: normal-weight asthmatics, overweight asthmatics, and asthmatics with metabolic syndrome. After an analysis of the results, it was observed that among asthmatics with overweight and metabolic syndrome, the serum lipids were significantly different from normal-weight asthmatics and the control group. In addition, in those with metabolic syndrome, the values of total cholesterol, triglycerides, and LDL-C were significantly higher compared to obese patients [21].

In those with metabolic syndrome, a significant difference was observed in LDL-C and the sex of the patients. Men had an LDL-C of 256.5 ± 69.60 mg/dL, while women had an LDL-C of 154.65 ± 78.86 mg/dL. Regarding LDL-C, such differences were also observed among overweight patients, with women having an LDL-C of 137.94 ± 52.07 mg/dL and men having an LDL-C of 176.78 ± 76.02 mg/dL. Between the group of normal-weight asthmatic patients and the control group, differences were observed in TG, with values of 109.65 ± 33.75 mg/dL vs. 92.63 ± 35.37 mg/dL, but LDL-C also had a significantly lower value than the control group, 68.76 ± 32 mg/dL vs. 85.84 ± 39.39 mg/dL. Therefore, the most significant changes in serum lipids were in overweight asthmatics and those with metabolic syndrome. To avoid long-term complications and repeated exacerbations, it is recommended to monitor the serum lipids and to adopt a lifestyle approach leading in time to the resolution or at least the improvement of these syndromes [21].

Excess fat tissue leads to chronic inflammation, which impedes easy asthma control. Moreover, insulin resistance and hypertension, components of the syndrome, are thought to be associated with the condition [31]. Following a comprehensive systematic review and meta-analysis of observational studies, Nahid Karamzad et al. concluded that metabolic syndrome is more common in asthma patients, but without statistical significance. Therefore, systemic and airway inflammation could be related to inflammatory cytokines secreted in adipose tissue [32].

Between January 2004 and December 2015, 76,368 people were enrolled in a longitudinal observational study to assess the correlation between asthma and metabolic syndrome in Chinese adults. At the first assessment, an important criteria was that people did not have a diagnosis of bronchial asthma. After clinical, paraclinical, and questionnaire evaluation, 4064 participants were excluded. Eligible subjects were investigated to determine the diagnosis of metabolic syndrome. Factors considered were obesity or overweight, hypertension, hyperglycemia, elevated triglyceride values, or low HDL-cholesterol values. During the study, 90 people were diagnosed with bronchial asthma: 38 women and 52 men. It could be observed that in the case of older age women, hypertriglyceridemia, increased body weight, and a history of respiratory disease were factors influencing the onset of asthma. In men, a history of respiratory disease, smoking, alcohol, and diastolic hypertension played this role. Finally, after a detailed analysis, it was concluded that overweight and obesity among women were the only significant risk factors that led to the development of asthma and not metabolic syndrome [29].

In 2021, Mohammad Emami-Ardestani and Ghazaleh Sajadi published an article discussing the same topic. They conducted a descriptive cross-sectional study in which they included 200 patients diagnosed with asthma. They grouped the patients based on the severity of their disease: intermittent, mild, moderate, and severe. They were assessed for pulmonary function and blood pressure, blood samples were taken, and they were measured and weighed to establish the diagnosis of metabolic syndrome. Of the 200 participants, 77 were diagnosed with metabolic syndrome, but no correlation was determined between this diagnosis and asthma severity categories. However, an increased prevalence of metabolic syndrome was observed among asthmatics compared to healthy individuals, suggesting that there may be an association between the two, but further studies are needed [33].

Abdellah H.K Ali also mentions in a study that included 320 patients with bronchial asthma that metabolic syndrome seems to be more common in asthmatics. The patients underwent the same type of evaluation as in the studies mentioned above, and it was observed that among the most common factors were obesity with abdominal disposition in 40% of the participants, low HDL-cholesterol in the same percentage, and hyperglycemia in 27.5% of the patients. Of the 320 participants, 57.5% were diagnosed with metabolic syndrome, which reinforces the idea that metabolic syndrome is common among asthmatic patients [34].

### 4.6. Nutrition

In addition to the basic drug treatment used to control asthma, patients also need to make lifestyle changes. The link between serum lipids and asthma raises the question of whether diet is important in these patients [2]. It is well known that excess fat tissue stimulates the appearance of pro-inflammatory cytokines, which leads to the worsening of the disease and an increase in the number of asthma exacerbations [35].

Western-style diets, high in animal products and saturated fats, are thought to lead to increased inflammation, particularly in the airways, leading over time to impaired lung function and worsening asthma [2,36]. Dairy consumption has been positively correlated in several studies with the risk of developing asthma, but there is no clear data on this yet [2].

Even though fats can be harmful, it appears that olive oil consumption has been associated in numerous studies with a decreased risk of the disease, offsetting the harmful effect on symptoms of butter consumption [35]. As for fish oil, rich in omega-3, studies are not very clear, but the balance tends to lean towards the beneficial effects of regular consumption.

The greatest impact on the development of asthma seems to be the consumption of fruit and vegetables. Several studies show that a predominantly vegan diet is associated with better control of the disease and a decrease in the number of pro-inflammatory cytokines. Antioxidants in fruits and vegetables, such as flavonoids, vitamin E, and vitamin C, decrease inflammation in the airways and improve lung function [2,37]. In addition, predominantly plant-based diets bring with them increased fiber intake. Fiber influences processes in the gut microbiome, leading to the formation of short-chain fatty acids such as butyrate. These fatty acids are thought to decrease the effect of pro-inflammatory cytokines, thus decreasing the exaggerated response to allergens and, therefore, leading to better disease control [2,38]. In addition, fiber, by creating an extra barrier in the gut, hinders protein absorption, with an additional anti-allergic effect [2].

Therefore, the most successful diets for asthma patients seem to be those that increase the consumption of fruit, vegetables, and high-fiber foods because all of these food groups are rich in vitamins, antioxidants, and minerals [35].

Asthma patients need to be careful not only about the diet they adopt, but also when they eat meals. Meal times can have a major influence on serum lipids, with numerous studies correlating feeding times with metabolic impairment, obesity, and associated pathologies [39]. An imbalance between the circadian rhythm of metabolic rate and lunch can lead, over time, to impaired lipid profiles. In a study published in 2019, it was observed that just 100 kcal of extra fat consumed at night can lead to an increase in LDL cholesterol by 2.98 mg/dL. At the opposite end of the spectrum, eating high-calorie foods in the morning not only lowers cholesterol, but also helps with weight loss. This is explained by the fact that the metabolic system produces more cholesterol at night, and an increasing intake at this time would only raise cholesterol levels [40].

Another aspect that seems to influence serum lipids is time-restricted eating. In a systematic review that included several studies on time-restricted eating and its effect on human health, it is mentioned that the HDL-cholesterol level decreased after a period of time when the participants were fasting. This is mostly because of the adipokine level, which increases during fasting [41].

In another study, Wilkinson et al. mention that LDL cholesterol and non-HDL cholesterol levels were reduced after 12 weeks of 10/14 time-restricted eating in 19 adults diagnosed with metabolic syndrome before the inclusion in the study [42]. Therefore, it is not only the quality and quantity of food that is important, but it seems that the timing of eating also influences serum lipid levels and the occurrence of obesity, metabolic syndrome, and other associated pathologies.

All of the above information are summarized in Figure 5.

### 4.7. Murine Model

Yan-Fan Yeh and collaborators wanted to find out whether a diet high in cholesterol and, thus, diet-induced hypercholesterolemia can affect lung function in murine asthma models. Thus, the researchers offered mice different diets: the control group received a diet with 0.02% cholesterol, and the target groups’ diets were supplemented with 1% or 2% cholesterol. After one month, the murine models were sensitized with intraperitoneal ovalbumin, and after 2 weeks, sensitization was supplemented with inhaled ovalbumin or saline. In mice receiving a high-cholesterol diet, serum cholesterol values increased during the study. In this experiment, an increase in the number of eosinophils in the bronchoalveolar lavage fluid and in pro-inflammatory cytokines such as IL-5 was observed in mice on a high-cholesterol diet [43].

Expanding on the discussion of potential statin therapy in patients with bronchial asthma, it can be mentioned that beneficial effects have also been observed in murine model studies. Fatih Fırıncı and co-workers concluded that atorvastatin, through its anti-inflammatory role, leads to a decrease in IL-4 and IL-5 cytokines [44]. In addition, atorvastatin and simvastatin appear to decrease the number of eosinophils and macrophages in bronchoalveolar lavage fluid and improve dyslipidemia even in obese subjects [44,45,46]. In another study focusing on the same subject, rosuvastatin was found to decrease airway inflammation in murine subjects due to decreased oxidative stress, white blood cell counts, and pro-inflammatory cytokines in addition to its hypolipidemic role [47].

All of the above results are summarized in Table 3, in which a statistically significant correlation between serum lipids and different parameters can be observed.

## 5. Conclusions

Asthma is a respiratory disease caused by chronic airway inflammation associated with various risk factors. Among these risk factors is abnormal serum lipids, more specifically, dyslipidemia. Studies show a positive correlation between total cholesterol, triglyceride, and LDL-cholesterol levels and asthma and a negative correlation between HDL-cholesterol and asthma. Increased cholesterol values would lead to the stimulation of pro-inflammatory processes and the secretion of cytokines involved in these processes. Because of this correlation, it is questionable whether patients would benefit additionally if, in addition to basic therapy, they were given lipid-lowering medication or adopted a predominantly plant-based diet. Further studies are needed on these issues.

The current study has several limitations. The analysis was not stratified according to asthma severity or levels of all serum lipids in all included studies. In addition, the methods of assessing serum lipids and lung function varied across studies, which could lead to different results. Furthermore, the diagnosis of asthma was made by different methods in each study. The publication period of the articles was limited to the last seven years because we wanted to evaluate only articles that were not included in the previous systematic review. In addition, other reviews, meta-analyses, and systematic review articles were not included in this systematic review. For this systematic review, we chose to limit ourselves only to studies that included adults over 18 years of age, excluding children and the elderly.

## Figures and Tables

**Figure 1 nutrients-16-02070-f001:**
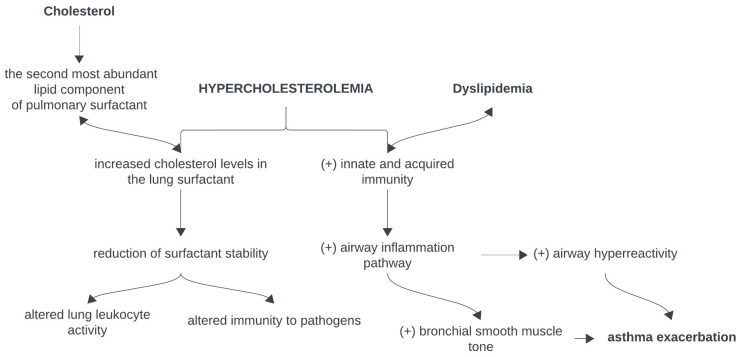
Association between hypercholesterolemia and asthma. “(+)”—“stimulation of” [6,7,8].

**Figure 2 nutrients-16-02070-f002:**
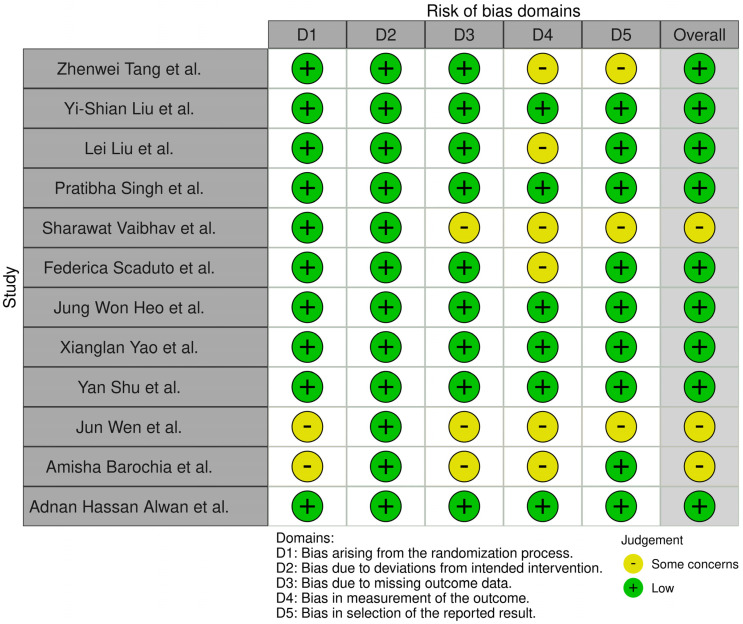
Methodological quality summary: review authors’ judgement about each methodological quality item for each included study [10,11,12,13,14,15,16,17,18,19,20,21].

**Figure 3 nutrients-16-02070-f003:**
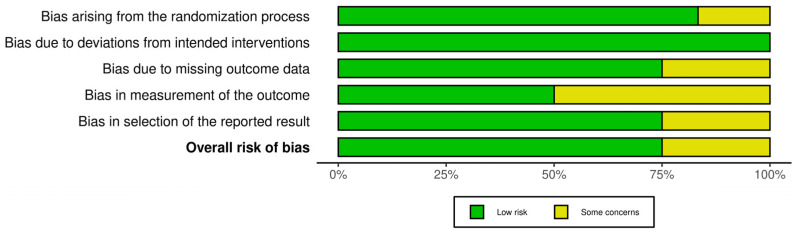
Results that represent the different types of biases (domains) across the studies included in the systematic review.

**Figure 4 nutrients-16-02070-f004:**
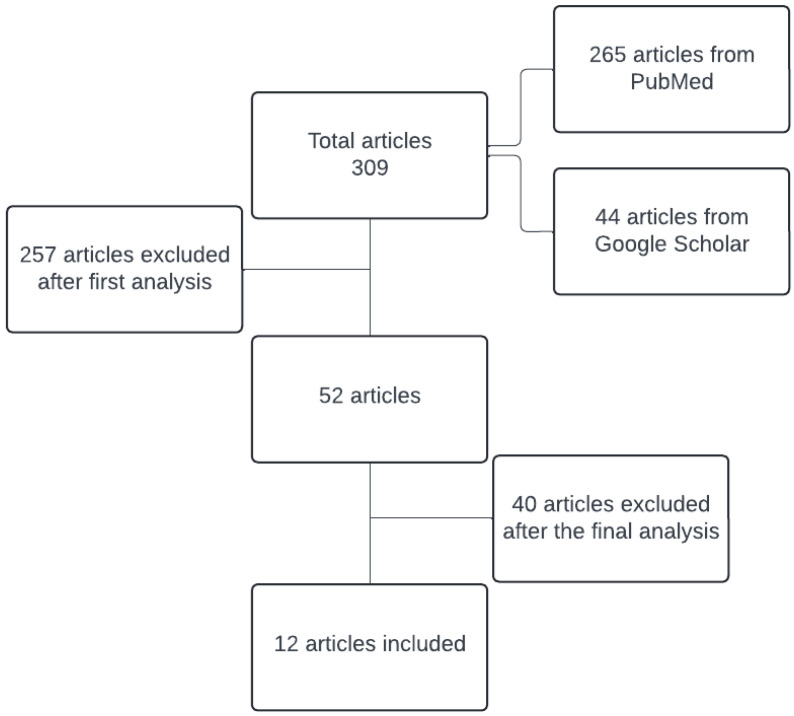
Exclusion steps for articles included in the systematic review.

**Figure 5 nutrients-16-02070-f005:**
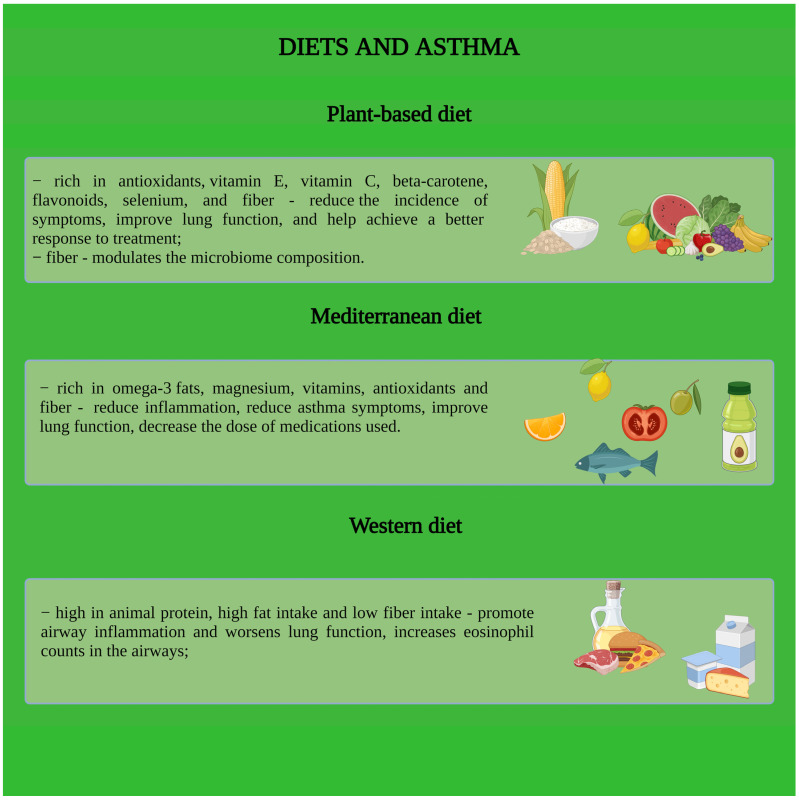
Nutritional aspects in asthmatic patients.

**Table 1 nutrients-16-02070-t001:** Electronic search strategy on PubMed with PubMed Advanced Search Builder.

Filters Applied: Humans, Adults: 19+ Years, from 2017 to 2024		
Terms	Query	Results
(high-density lipoprotein) AND (asthma)	((“lipoproteins, hdl”[MeSH Terms] OR (“lipoproteins”[All Fields] AND “hdl”[All Fields]) OR “hdl lipoproteins”[All Fields] OR (“high”[All Fields] AND “density”[All Fields] AND “lipoprotein”[All Fields]) OR “high density lipoprotein”[All Fields]) AND (“asthma”[MeSH Terms] OR “asthma”[All Fields] OR “asthmas”[All Fields] OR “asthma s”[All Fields])) AND ((humans[Filter]) AND (alladult[Filter]) AND (2017:2024[pdat]))	23
(low-density lipoprotein) AND (asthma)	((“lipoproteins, ldl”[MeSH Terms] OR (“lipoproteins”[All Fields] AND “ldl”[All Fields]) OR “ldl lipoproteins”[All Fields] OR (“low”[All Fields] AND “density”[All Fields] AND “lipoprotein”[All Fields]) OR “low density lipoprotein”[All Fields]) AND (“asthma”[MeSH Terms] OR “asthma”[All Fields] OR “asthmas”[All Fields] OR “asthma s”[All Fields])) AND ((humans[Filter]) AND (alladult[Filter]) AND (2017:2024[pdat]))	26
(cholesterol) AND (asthma)	((“cholesterol”[MeSH Terms] OR “cholesterol”[All Fields] OR “cholesterol s”[All Fields] OR “cholesterole”[All Fields] OR “cholesterols”[All Fields]) AND (“asthma”[MeSH Terms] OR “asthma”[All Fields] OR “asthmas”[All Fields] OR “asthma s”[All Fields])) AND ((humans[Filter]) AND (alladult[Filter]) AND (2017:2024[pdat]))	81
(lipid profile) AND (asthma)	((“lipid s”[All Fields] OR “lipidate”[All Fields] OR “lipidated”[All Fields] OR “lipidates”[All Fields] OR “lipidation”[All Fields] OR “lipidations”[All Fields] OR “lipide”[All Fields] OR “lipides”[All Fields] OR “lipidic”[All Fields] OR “lipids”[MeSH Terms] OR “lipids”[All Fields] OR “lipid”[All Fields]) AND (“profile”[All Fields] OR “profiled”[All Fields] OR “profiler”[All Fields] OR “profilers”[All Fields] OR “profiles”[All Fields] OR “profiling”[All Fields] OR “profilings”[All Fields]) AND (“asthma”[MeSH Terms] OR “asthma”[All Fields] OR “asthmas”[All Fields] OR “asthma s”[All Fields])) AND ((humans[Filter]) AND (alladult[Filter]) AND (2017:2024[pdat]))	43
(serum lipid) AND (asthma)	((“serum”[MeSH Terms] OR “serum”[All Fields] OR “serums”[All Fields] OR “serum s”[All Fields] OR “serumal”[All Fields]) AND (“lipid s”[All Fields] OR “lipidate”[All Fields] OR “lipidated”[All Fields] OR “lipidates”[All Fields] OR “lipidation”[All Fields] OR “lipidations”[All Fields] OR “lipide”[All Fields] OR “lipides”[All Fields] OR “lipidic”[All Fields] OR “lipids”[MeSH Terms] OR “lipids”[All Fields] OR “lipid”[All Fields]) AND (“asthma”[MeSH Terms] OR “asthma”[All Fields] OR “asthmas”[All Fields] OR “asthma s”[All Fields])) AND ((humans[Filter]) AND (alladult[Filter]) AND (2017:2024[pdat]))	72
(triglyceride) AND (asthma)	((“triglycerid”[All Fields] OR “triglycerides”[MeSH Terms] OR “triglycerides”[All Fields] OR “triglyceride”[All Fields] OR “triglycerids”[All Fields]) AND (“asthma”[MeSH Terms] OR “asthma”[All Fields] OR “asthmas”[All Fields] OR “asthma s”[All Fields])) AND ((humans[Filter]) AND (alladult[Filter]) AND (2017:2024[pdat]))	20

**Table 2 nutrients-16-02070-t002:** Included articles.

First Author, Year	Country	Study Duration	Study Design	Number of Participants	Age and Sex of Participants
Barochia Amisha V. et al. (2017) [20]	USA	1999–2015	Cohort study	n = 333	NN—mean age 32 ± 12.4 years, 70% women and 30% men. AN—mean age 34 ± 12.4 years, 49% women and 51% men AA—mean age 37.3 ± 14.2 years, 64% women and 35% men.
Adan Hassan Alwan et al. (2018) [21]	Iraq	2010–2012	Case–control	n = 348	Mean age 34.34 ± 11.58 years, 61% women and 39% men
Scaduto Federica et al. (2018) [7]	Italy	2018	Observational	n = 80	Age >18 years, 52.5% women and 47.5% men
Won Heo Jung et al. (2020) [16]	South Korea	2019	Retrospective and observational	n = 167	PFT—mean age 52.2 ± 18.8 years, 24.7% male PD—mean age 58 ± 18.1 years, 27% male.
Zhenwei Tang et al. (2022) [10]	England, Scotland, Wales	2006–2010	Observational	n = 502.505	Mean age 56.5 years, 52.7% women
Liu Lei et al. (2022) [12]	China	2015–2019	Observational	n = 477	NL—mean age 42.2 ± 13.73, 70.3% women DL—mean age 50.14 ± 12.61, 58.3% women
Yan Shu and Wei Wang (2022) [18]	USA	2004–2006	Cross-sectional	n = 1013	Mean age 52.5 years, 44.91% men.
Vaibhav Sharawat et al. (2023) [14]	India	2023	Observational	n = 77	Age range: 18–60 years 41.6% women and 58.4% men
Yao Xianglan et al. (2023) [17]	USA	1999–2016	Longitudinal study	n = 300	A: 39 ± 15.2 years, 66% women NA: 34.5 ± 13.6 years, 55% women
Wen Jun et al. (2023) [19]	USA	2011–2018	Cross-sectional	n = 2.544	Age > 18 years, 58.96% women
Yi-Shian Liu et al. (2024) [11]	Taiwan	2012–2019	Bidirectional two-sample Mendelian randomization study	n = 24.853	Mean age 48.8 years, 49.8% women
Singh Pratibha et al. (2024) [13]	North India	2021-2022	Observational prospective cohort	n = 107	Mean age 32.97 ± 11.75 years, 51.40% women

NN, non-asthmatic and non-atopic status; AN, asthmatic non-atopic status; AA, asthmatic atopic status; PFT, pulmonary function test proven asthma; PD, physician-diagnosed asthma; NL, normal serum lipids patient; DL, dyslipidemic patient; USA, United States of America; n, number of participants.

**Table 3 nutrients-16-02070-t003:** Statistically significant correlations between different studied parameters in asthmatic patients.

Author, Year	Studied Parameters		Statistical Significance	Explanations
Adnan Hassan Alwan et al., 2018 [21]	normal weight overweight metabolic syndrome	Serum cholesterol	*p* = 0.42 *p* < 0.001 *p* < 0.001	Lack of physical activity and unhealthy eating habits lead to obesity and metabolic syndrome. Hypercholesterolemia stimulates inflammation in asthma.
		Serum triglyceride	*p* = 0.0125 *p* < 0.001 *p* < 0.001	
		HDL-C	*p* = 0.1215 *p* < 0.001 *p* < 0.001	
		LDL-C	*p* = 0.0179 *p* < 0.001 *p* < 0.001	
Yan Shu et al., 2022 [18]	Total cholesterol	Global sleep score (subjective sleep quality, sleep latency, sleep duration, habitual sleep efficiency, sleep disturbance, use of sleep medication, and daytime dysfunction)	*p* = −0.062	HDL-C has anti-inflammatory effects and exerts antioxidant activity.
	Triglycerides		*p* = 0.136	
	HDL-C		*p* = −0.199	
	LDL-C		*p* = −0.018	
	Total/HDL-C ratio		*p* = 0.154	
Jung Won Heo et al., 2021 [16]	TC	FeNO (Asthma confirmed via 1. pulmonary function test or 2. physician-diagnosis)	*p* = 0.03 *p* = 0.20	Metabolic syndrome and dyslipidemia are known to involve systematic inflammation, and are associated with asthma development and progression. Association between dyslipidemia and eosinophilic airway inflammation represented by FeNO.
	HDL		*p* = 0.42 *p* = 0.78	
	LDL		*p* = 0.01 *p* = 0.05	
	TG		*p* = 0.49 *p* = 0.05	
	ApoA		*p* = 0.34 *p* = 0.84	
	ApoB		*p* = 0.02 *p* = 0.25	
Amisha V. Barochia et al., 2017 [20]	TC	Eosinophil counts (1. non-asthmatic & 2. asthmatic)	*p* = 0.53 *p* = 0.92	HDL has antioxidant, anti-thrombotic, and anti-inflammatory functions—these properties may contribute to the negative correlation with blood eosinophil counts.
	Triglycerides		*p* = 0.99 *p* = 0.01	
	LDL-C		*p* = 0.43 *p* = 0.76	
	HDL-c–HDL particles (Total HDL NMR and large HDL NMR)		*p* = 0.41 *p* = 0.04 (Total HDL: *p* = 0.01)	
Zhenwei Tang et al., 2022 [10]	TG	Asthmatic status	*p* < 0.001, β = −0.005	Serum lipid dysregulation can contribute to the onset of atopic diseases.
	LDL		*p* < 0.001, β = −0.003	
	TC		*p* < 0.001, β = −0.002	
	HDL		*p* < 0.001, β = 0.004	
Lei Liu et al., 2023 [12]	FEV1 (% predicted)	Normal lipidemia vs. dyslipidemia	*p* = 0.012	Dyslipidemia is independently linked to pulmonary function and sensitization.
	FEV1/FVC (%)	Normal lipidemia vs. dyslipidemia	*p* = 0.004	
	Severe asthma exacerbation	Normal lipidemia vs. dyslipidemia	*p* = 0.022	
Pratibha Singh et al., 2023 [13]	TC	Controlled vs. uncontrolled asthma	*p* = 0.569	Concomitant asthma, airway blockage, and, in particular, airway resistance are linked with high levels of LDL, while high levels of HDL are linked with lower bronchial hyperresponsiveness and improved specific airway resistance.
	TG		*p* = 0.704	
	LDL		*p* = 0.030	
	HDL		*p* = 0.636	
	TC/HDL		*p* = 0.047	
Jun Wen et al., 2023 [19]	TC	Eosinophil counts	*p* = 0.5402	Metabolic syndrome and dyslipidemia contribute to the development of an asthmatic pro-inflammatory state. Eosinophilic airway inflammation has emerged as a defining characteristic of one severe form of asthma.
	LDL-C		*p* = 0.7836	
	HDL-C		*p* = 0.004	
	Triglyceride		*p* = 0.5443	
Yi-Shian Liu et al., 2023 [11]	TC	Asthma status	*p* = 0.001	Cholesterol is an important component of pulmonary surfactant. The synthesis of lung cholesterol is primarily derived from serum lipoproteins. Moreover, high-density lipoproteins play a central role in the acceleration of pulmonary surfactant production, and higher levels of HDL cholesterol were negatively associated with airway resistance and might reduce the risk of bronchial responsiveness.
	LDL-C		*p* = 0.001	
	HDL-C		*p* = 0.01 (inversely correlated)	

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
