# Peer review of "Association between Serum Lipids and Asthma in Adults—A Systematic Review"

_nutrients, 2024, doi:10.3390/nu16132070_

Round 1

Reviewer 1 Report (Previous Reviewer 1)

Comments and Suggestions for Authors

Maștaleru et al. aim to identify the possible association between asthma patients and their plasma lipids including HDL-cholesterol, LDL-cholesterol, total cholesterol, and triglyceride in adults. A systematic methodology has been adopted in collecting articles, excluding articles and selecting articles and focused review was made attaining the above objective. The results of this study should provide research gaps for conducting more in vivo experiments to prove the elucidated correlations. Therefore I recommend acceptance of this review article for publication after addressing the following minor comments:

1.        To make this review article more attractive, it is important to add some key figures (at least 2 or 3) from the reviewed articles after obtaining copyright from the respective publishers.

2.        At the end of discussion, a schematic diagram detailing the parameters chosen for establishing the intended association, the parameters that are actually correlating and the plausible mechanism of correlation should be included.

Comments on the Quality of English Language

Minor editing of English language required

Author Response

Dear reviewer,

Thank you for all your comments.

Maștaleru et al. aim to identify the possible association between asthma patients and their plasma lipids including HDL-cholesterol, LDL-cholesterol, total cholesterol, and triglyceride in adults. A systematic methodology has been adopted in collecting articles, excluding articles and selecting articles and focused review was made attaining the above objective. The results of this study should provide research gaps for conducting more in vivo experiments to prove the elucidated correlations. Therefore I recommend acceptance of this review article for publication after addressing the following minor comments:

  1. To make this review article more attractive, it is important to add some key figures (at least 2 or 3) from the reviewed articles after obtaining copyright from the respective publishers.

We have added more figures/diagrams that explain better the correlation between the lipid parameters and the studied risk factors in asthmatic patients. Thank you for your idea! We also have in our article a graphical abstract that marks the associations observed in our study. In addition, figure 1 includes plausible mechanisms between the pulmonary function and lipid profile in asthmatic patients.

At the end of discussion, a schematic diagram detailing the parameters chosen for establishing the intended association, the parameters that are actually correlating and the plausible mechanism of correlation should be included.

Thank you for your suggestion. We have added a table in the manuscript.

Hope we have touched all the points you asked us to change.

If there are any other changes you consider we should make, please let us know.

Yours sincerely,

All the authors

Reviewer 2 Report (Previous Reviewer 2)

Comments and Suggestions for Authors

1. In the title of the paper The Authors stated "serum lipids" while in the abstract and the entire text there is "plasma lipids". The Authors should know about the differences between plasma and serum and verify and decide whether it is serum or plasma that was considered.

2. The exclusion/inclusion criteria should be described in more details. FOr example, there is no information whether the subjects were characterized by the presence of any disease or medication use that in fact may affect plasma/serum?lipids or induce asthma, such as ASA or a beta-blockers, etc. Also, corticosteroids used for asthma may increase total cholesterol, therefore, the study  should be performed carefully and include all the issues that may give fals-positive results.

3. Including pediatric population or elderly may be problematic to provide, in the final, reliable results. Are there any other limitations? 

Comments on the Quality of English Language

minor English correction is needed

Author Response

Dear reviewer,

Thank you for all your comments.

  1. In the title of the paper The Authors stated "serum lipids" while in the abstract and the entire text there is "plasma lipids". The Authors should know about the differences between plasma and serum and verify and decide whether it is serum or plasma that was considered.

We have made this correction. Thank you very much for your remark!

2. The exclusion/inclusion criteria should be described in more details. FOr example, there is no information whether the subjects were characterized by the presence of any disease or medication use that in fact may affect plasma/serum?lipids or induce asthma, such as ASA or a beta-blockers, etc. Also, corticosteroids used for asthma may increase total cholesterol, therefore, the study  should be performed carefully and include all the issues that may give fals-positive results.

We have added a more extensive description of the inclusion and exclusion criteria in the text. Moreover, we have added more details regarding the treatment used for the asthmatic patients included in our systematic review before Table 2. We consider your suggestion very useful for our paper! Thank you! 

3. Including pediatric population or elderly may be problematic to provide, in the final, reliable results. Are there any other limitations? 

We have added a more detailed paragraph regarding our study limitations.

Hope we have touched all the points you asked us to change.

If there are any other changes you consider we should make, please let us know.

Yours sincerely,

All the authors

Reviewer 3 Report (Previous Reviewer 3)

Comments and Suggestions for Authors

Comments:

   The manuscript describes "Association between serum lipids and asthma in adults – a systematic review.” Asthma is a syndrome common in adults and children characterized by airflow obstruction caused by airway inflammation. Lipid metabolism plays an important role in exacerbating and reducing lung inflammation. The purpose of this study was to identify any association between patients diagnosed with asthma and their lipid profiles. The results demonstrated that triglycerides, total cholesterol, and low-density lipoprotein cholesterol were positively correlated with asthma. Furthermore, in terms of nutrition in asthmatics, the greatest influence on the development of the disease appears to be the intake of fruits and vegetables. Multiple studies have shown that a predominantly vegan diet improves disease control and reduces pro-inflammatory cytokines. Studies have shown that elevated cholesterol values ​​stimulate pro-inflammatory processes and the secretion of cytokines involved in these processes. The most successful diets for people with asthma appear to increase their intake of fruits, vegetables, and high-fiber foods, which are rich in vitamins, antioxidants, and minerals, but several points need clarification.

 Comment:

1. The abstract of this article does not fit this magazine.

 2. Nutrition in patients with asthma appears to have the greatest impact on the development of the disease, the intake of fruits and vegetables. Multiple studies have shown that a predominantly vegan diet improves disease control and reduces pro-inflammatory cytokines. However, the authors state that it is questionable whether additional benefits would come from patients receiving lipid-lowering drugs or adopting a plant-based diet in addition to basic treatment. The authors should independently analyze the graphs and their results.

 3. Nutrition in patients with asthma appears to have the greatest impact on the development of the disease, the intake of fruits and vegetables. Timing of nutrient administration Authors should analyze their results

4. The title of the chart should be a detailed description rather than a brief description.

Comments on the Quality of English Language

Minor editing of English language required

Author Response

Dear reviewer, 

Thank you for your comments.

The manuscript describes "Association between serum lipids and asthma in adults – a systematic review.” Asthma is a syndrome common in adults and children characterized by airflow obstruction caused by airway inflammation. Lipid metabolism plays an important role in exacerbating and reducing lung inflammation. The purpose of this study was to identify any association between patients diagnosed with asthma and their lipid profiles. The results demonstrated that triglycerides, total cholesterol, and low-density lipoprotein cholesterol were positively correlated with asthma. Furthermore, in terms of nutrition in asthmatics, the greatest influence on the development of the disease appears to be the intake of fruits and vegetables. Multiple studies have shown that a predominantly vegan diet improves disease control and reduces pro-inflammatory cytokines. Studies have shown that elevated cholesterol values ​​stimulate pro-inflammatory processes and the secretion of cytokines involved in these processes. The most successful diets for people with asthma appear to increase their intake of fruits, vegetables, and high-fiber foods, which are rich in vitamins, antioxidants, and minerals, but several points need clarification.

Comment:

  1. The abstract of this article does not fit this magazine.

We have reevaluated our abstract and changed it. Thank you! The special issue in which we have submitted our article is entitled “Diet, Asthma and Respiratory Health”, thus we have considered it is suitable for this journal.

  1. Nutrition in patients with asthma appears to have the greatest impact on the development of the disease, the intake of fruits and vegetables. Multiple studies have shown that a predominantly vegan diet improves disease control and reduces pro-inflammatory cytokines. However, the authors state that it is questionable whether additional benefits would come from patients receiving lipid-lowering drugs or adopting a plant-based diet in addition to basic treatment. The authors should independently analyze the graphs and their results.

We have added a table in our study that includes the correlations between the lipid profile and the risk factors in asthmatic patients. Moreover, we have added a figure that highlights the correlations between types of diets in patients with asthma.

  1. Nutrition in patients with asthma appears to have the greatest impact on the development of the disease, the intake of fruits and vegetables. Timing of nutrient administration Authors should analyze their results

We have added more details regarding the timing of nutrient administration. Thank you!

  1. The title of the chart should be a detailed description rather than a brief description.

We have added a more detailed description of the chart.

Hope we have touched all the points you asked us to change.

If there are any other changes you consider we should make, please let us know.

Yours sincerely,

All the authors

Round 2

Reviewer 2 Report (Previous Reviewer 2)

Comments and Suggestions for Authors

The Authors have considered all the suggested corrections. In my opinion, the paper is suitable for publication

Reviewer 3 Report (Previous Reviewer 3)

Comments and Suggestions for Authors

Accepted

Comments on the Quality of English Language

Minor editing of English language required

This manuscript is a resubmission of an earlier submission. The following is a list of the peer review reports and author responses from that submission.

Round 1

Reviewer 1 Report

Comments and Suggestions for Authors

Mastaleru et al. present a systematic review on association between lipid profile and asthma in adults. It is an important and timely review that is well-approached, especially in the design and discussion part. However, the following issues need to be addressed for a possible publication in Nutrients.

1.        The introduction is too short and should be elaborated further.

2.        The objectives should be placed as the last sentence of the introduction.

3.        Two paragraphs in section 2.2 should be combined as one paragraph.

4.        The captions of Table 1 and 2 as well as Figure 1 should be more descriptive.

5.        Table 2 – “Irak” should be “Iraq”.

6.        Table 2 – how the number of participants (n) is in fractions?

7.        The details presented in Table 2 should also be consolidated in the text by grouping different categories.

8.        There are no key figures provided in this review. The authors should pick some reported key figures that address the association between lipid profile and asthma to be reproduced in this article after obtaining copyright permission.

9.        A schematic diagram detailing the scientific scope of this review article with the parameters reviewed in this article should be included.

10.     A schematic diagram illustrating the mechanistic association between lipid profile and asthma should be provided.

11.     The limitations in section 6 should be moved to the conclusion section.

Comments on the Quality of English Language

Minor editing of English language required

Author Response

Dear reviewer,

Thank you for all your comments.

  1. We have added more details and made the introduction section enlarged. Thank you!
  2. We have placed the objectives in the last paragraph of the Introduction section.
  3. Thank you for the suggestion! We have made this requirement.
  4. We have added more descriptive details. Thank you!
  5. We have made this correction. Thank you for your observation.
  6. In Table 2, we have added details from all the included articles and taken the fractions from the cited research.
  7. We have added a more extensive description in the text, before Table 2, and we have listed the different criteria groups.
  8. We have added two more figures, one a graphical abstract and the other representing the correlation between hypercholesterolemia and asthma.
  9. We have moved the limitations section into the conclusion section.

Hope we have touched all the points you asked us to change.

If there are any other changes you consider we should make, please let us know.

Yours sincerely,

All the authors

Reviewer 2 Report

Comments and Suggestions for Authors

The Authors presented an analysis of the literature aimed at describing any relationship between the incidence of asthma and the lipid profile. Although the title sounds interesting, the text should be heavily rewritten.

1. The Authors collected over 300 papers, but only 12 were selected for further analysis. I understand that only these articles met the inclusion criteria, but it would be nice to read what was the exact reason why the Authors excluded the papers (some examples).

2. the results only contain a table showing what type of articles were included. However, in my opinion, this data should be described in more detail (some parts of the discussion should be transferred to the results section)

3. The discussion should be rewritten as it is written in a rather chaotic way. I'm a bit confused as to why the Authors included additional information about statins, blood eosinophils, etc? Are these subtitles in the discussion related to the papers analyzed ?

4. HDL is known for its proinflammatory activity, and this is due to its molecular size, as different sizes are thought to induce various effects.

5. Since there are various types of asthma (i.e., allergic, exercise-induced, etc), did the Authors verify whether the correlation between lipid profile and asthma is strictly dependent on the type of asthma? 

Comments on the Quality of English Language

minor changes are required

Author Response

Dear reviewer,

Thank you for all your comments.

  1. We have added more details regarding the inclusion and exclusion criteria. Thank you!
  2. We have added a more extensive description in the text, before Table 2.
  3. We have added the paragraph regarding statin treatment because we considered it to have an important correlation with lipid profile dysfunction. Moreover, we considered that this could increase the quality of the paper and the curiosity of the readers. Regarding blood eosinophils, we considered it important to mention the bond between HDL and the number of eosinophils due to the well-known asthma classification.
  4. We have added a paragraph in which we have detailed the correlations between the different types of asthma and the cholesterol fractions. Thank you very much for your comments!

Hope we have touched all the points you asked us to change.

If there are any other changes you consider we should make, please let us know.

Yours sincerely,

All the authors

Reviewer 3 Report

Comments and Suggestions for Authors

Comments:

   The manuscript describes " Association between lipid profile and asthma in adults – a systematic review”. Asthma is a syndrome common in adults and children characterized by airflow obstruction caused by inflammation of the airways. In recent years, studies have found that lipid metabolism plays an important role in both the aggravation of the disease and the reduction of lung inflammation. This study was to determine the association between patients diagnosed with asthma and their blood lipids. The final results found that triglycerides, total cholesterol, and low-density lipoprotein cholesterol (LDL-cholesterol) were positively correlated with asthma. In terms of nutrition for people with asthma, the biggest influence on the development of the disease appears to be the intake of fruits and vegetables. Elevated cholesterol values ​​stimulate pro-inflammatory processes and the secretion of cytokines involved in these processes. The most successful diet for people with asthma appears to increase their intake of fruits, vegetables, and high-fiber foods, but several points need clarification.

 Comment:

1. A total of 309 articles from both platforms were analyzed. What are the criteria for articles on both platforms to be duplicated and to exclude articles that do not meet the eligibility criteria? The 12 articles selected from the recognition article pool are not recognized enough.

 2. Results from experimental animal studies should be included in the article.

 3. The description of the Figure legend should be strengthened

Comments on the Quality of English Language

Minor editing of English language required

Author Response

Dear reviewer,

Thank you for all your comments.

  1. We have added more details regarding the inclusion and exclusion criteria. Thank you!
  2. We had as an inclusion criteria only articles that were conducted on humans. This is a very good idea and we will consider it for future research.
  3. We have added more details for the figure legends. Thank you!

Hope we have touched all the points you asked us to change.

If there are any other changes you consider we should make, please let us know.

Yours sincerely,

All the authors

Round 2

Reviewer 2 Report

Comments and Suggestions for Authors

The Authors answered all my concerns, therefore I suggest this paper can be published 

Author Response

Thank you very much!

Reviewer 3 Report

Comments and Suggestions for Authors

The author did not answer related questions

Comments on the Quality of English Language

Minor editing of English language required

Author Response

Dear reviewer,

Thank you for your comments.

We have searched for research that included asthma and lipid profile in mice and we have added a subheading with them in the discussion section. We have found them very useful. Thank you very much for your idea.

Hope we have touched all the points you asked us to change.

If there are any other changes you consider we should make, please let us know.

Yours sincerely,

All the authors

Round 3

Reviewer 3 Report

Comments and Suggestions for Authors

The article uploaded by the author does not seem to be this article, and the content cannot be seen.

Comments on the Quality of English Language

 Minor editing of English language required